**Data Availability Statement:** All data are available at https://snap.stanford.edu/data/loc-gowalla.html.

**Funding:** This research was supported by a grant from the MSIT (Ministry of Science and ICT), Korea, under the ICT Creative Consilience program

# Safe contact tracing for COVID-19: A method without privacy breach using functional encryption techniques based-on spatio-temporal trajectory data

**Wooil Kim, Hyubjin Lee, Yon Dohn Chung** *

Department of Computer Science & Engineering, Korea University, Seoul, Republic of Korea

* ydchung@korea.ac.kr

## Abstract

The COVID-19 pandemic has spread all over the globe. In the absence of a vaccine, a small number of countries have managed to control the diffusion of viruses by early detection and early quarantine. South Korea, one of the countries which have kept the epidemics well-controlled, has opened the infected patients' trajectory to the public. Such a reaction has been regarded as an effective method, however, serious privacy breach cases have been issued in South Korea. Furthermore, some suspected contacts have refused to take infection tests because they are afraid of being exposed. To solve this problem, we propose a privacy-preserving contact tracing method based on spatio-temporal trajectory which can be practically used in many quarantine systems. In addition, we develop a system to visualize the contact tracing workflow.

## Introduction

As of June 2020, the ongoing pandemic of coronavirus disease 2019, abbreviated as "COVID-19", has caused more than 7 million cases and 0.4 million deaths over 188 countries and territories [1]. Each country has reacted to prevent the epidemic spreads with their ways; Countries in the Schengen area have restricted free movement [2], and Sweden has remained open in an attempt to reach herd immunity [3]. Despite their hard work, only a few countries have kept well-controlled the epidemic spread. Fig 1 shows the numbers of new cases of COVID-19 over countries which are early detected or currently most highly infected.

After the outbreak in Wuhan, China, in December 2019 [4], the virus was first confirmed to have spread to South Korea on 20 January 2020 [5]. By 28 February, more than 2,000 confirmed cases were reported in South Korea [6], and the number of cases exponentially grew for a few days. However, as Fig 1 shows, the number of new cases in South Korea looks almost not increasing after March, contrary to other countries. South Korea's reaction to the COVID-19 is regarded as a successful method without any lockdowns [7, 8].

The detailed quarantine system of South Korea is described in Fig 2. If an infected patient is detected, the government puts the patient in quarantine and provides medical care **(1)**. At the

(IITP-2020-0-01819) supervised by the IITP (Institute for Information & communications Technology Planning & Evaluation) for WK, HL and YDC, and under the framework of international cooperation program managed by the National Research Foundation of Korea (NRF-2020K2A9A1A01095894) for WK and YDC.

**Competing interests:** The authors have declared that no competing interests exist.

same time, they investigate the patient's trajectory based on multiple data sources (i.e. card payment records, mobile GPS data, and self-reports) (**2**). With the trajectory, the government traces and tests people who have contacted with the infected patient (**4a**). Considering the undetected contacts, the trajectory is open to the public as an anonymized form (**3, 4b**). Then the people who might have contacted the patient get tests voluntarily or enforcedly (**5**). If the contacts are tested as positive (**6a**), the same loop is repeated and negative contacts are observed and recommended to be self-quarantined (**6b**). With this system, the government of South Korea reported the lowest number of new cases on 23 March [9]. However, the system also has caused serious privacy problems. Though patients' identifiers (e.g., name) are not open to the public, their privacy has been breached in some extents. The privacy leakage often occurs in the process of disclosing the trajectory to the public. In Fig 2, when disclosing the trajectory to the public (**3**), tracing the detected contacts (**4a**), and undetected contacts (**4b**) the privacy problem may occur because of the travel route data contain information about individuals' privacy [10]. Individuals might be afraid that their visits are open to the public. For example, some suspected contacts in South Korea have been afraid that their visits to a gay bar are disclosed [11].

Commercial [12] and academic [13] solutions have been released for COVID-19 contact tracing. Utilizing these applications and research, people can check whether they have been exposed to the risk of being infected or not. TraceTogether [12] provides secure contact tracing method based on Bluetooth, however, a privacy problem has been issued [14]. Also, He et al. [13] proposed a contact tracing query and its processing method, but the privacy leakage

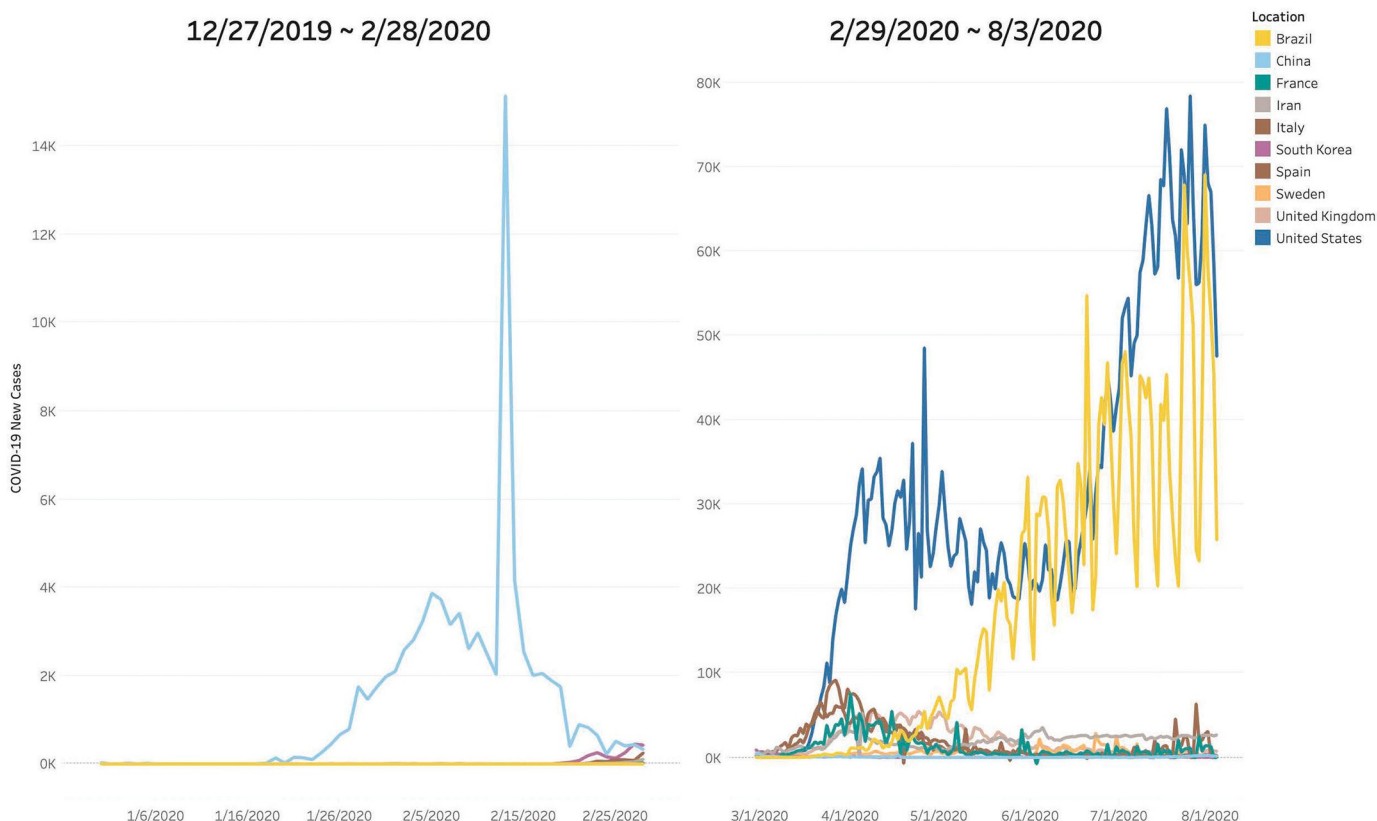

**Fig 1. COVID-19 new cases (3. August. 2020).** The number of COVID-19 new cases over countries (the early detected countries: China, South Korea, Sweden, and Italy; the most highly infected countries: Iran, Spain, France, United States, United Kingdom, and Brazil).

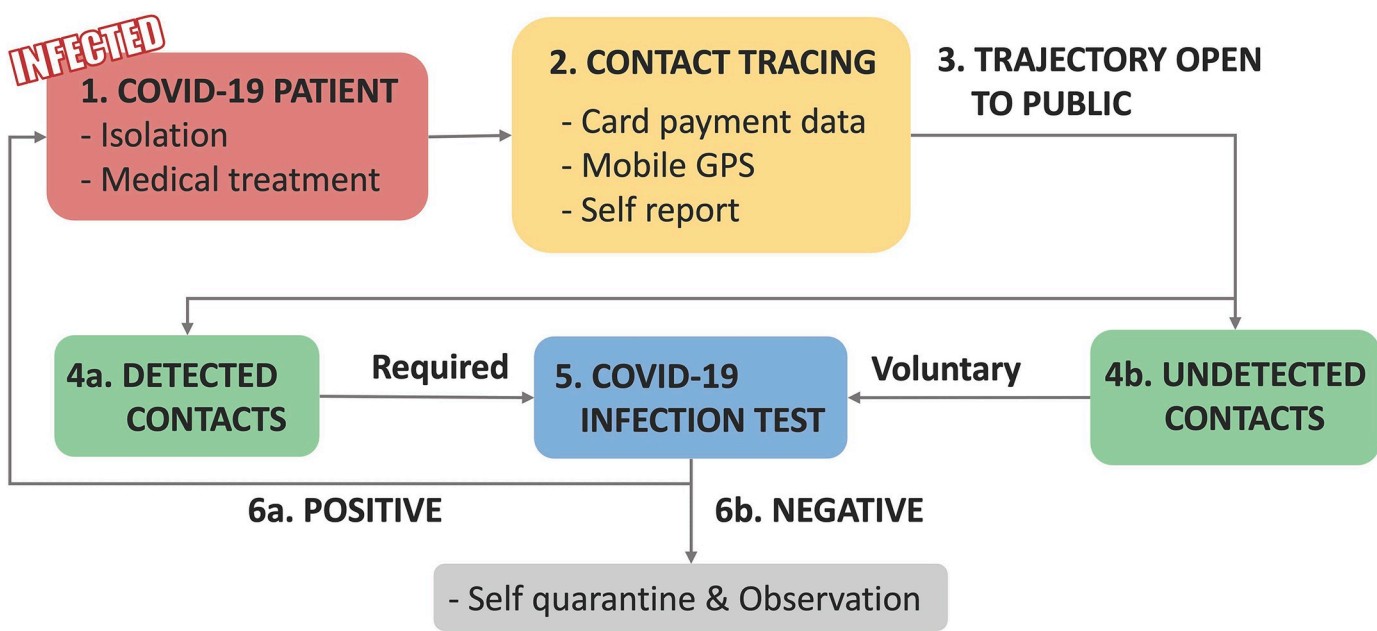

**Fig 2. Quarantine workflow of South Korea.**

problem is not considered. If privacy is not considered, people are reluctant to take COVID-19 infection tests, which makes the quarantine system less efficient. In the absence of vaccine, the importance of contact tracing will increase further. In this paper, we propose a privacy-preserving contact tracing method without privacy breaches. The contributions of this paper are as follows.

- We propose a privacy-preserving contact tracing method based on people's trajectory data.

  - We use the credit card payment data as the trajectory source (time and location). The trajectory data (i.e., person ID In the paper, we use the card number as the person ID., time and location) are stored in an encrypted form by using a functional encryption technique. Since the trajectory data are encrypted, the time and location privacy of people is preserved. However, the functional encryption scheme allows us to check a person visits the same place at the same time (with a specified time gap) with given confirmed cases' trajectory.

- We enhance the performance of the query processing via parallel processing and optimization techniques.

  - The functional encryption scheme requires a huge processing time, especially during the decryption step. The complexity is known as $O(n^2)$, where $n$ is the domain size. In order to tackle this, we propose a numeric decomposition method which divides numeric values of time and location into digits and encrypts them separately. In addition, we develop a multi-core, parallel processing algorithm with which the processing of contact tracing can be done significantly fast.

- We implement a system to visualize the workflow of contacts tracing process, where the suspected people are detected in real time represented as an infection graph.

  - The GUI system displays the progress of the contact tracing and shows the contamination relation among contacts and confirmed cases into dynamically changing graphs. Also, the

probability of contamination is displayed quantitatively for each contact, and the contamination sites are retrieved automatically.

## Literature review

### Contact tracing methods for COVID-19

Several approaches have been proposed for preventing the spread of COVID-19. Canetti et al. [15] argued that the GPS-based proximity detection methods involve an error inside a building and proposed the method based on Bluetooth. Bluetooth can be a solution for the indoor proximity detection, however, [15] needs an installed application in the devices to store and compute users' data. Singapore's TraceTogether application [12], also works based on Bluetooth. However, [14] showed that the users' infection status and the identifiers are revealed in TraceTogether. Other Bluetooth based methods [16, 17] were proposed considering the privacy. However, all such methods require an additional device or installed application to store and compute the data [15–17], or jeopardize the users' privacy [12]. He et al. propose a new contact tracing method based on trajectory similarity queries [13]. They developed a spatial index (SPT) to reduce I/O and data redundancy. However, the privacy breaches are not considered.

### Functional encryption

In a functional encryption system [18], a user is allowed to compute a function $F$ of a ciphertext $ct_x$ (i.e. encrypted version of plain text $x$) so that the results are available. In other words, a functionality $F$ of $ct_x$ returns the same result of the function of $x$. Clearly, a functional encryption system guarantees a plaintext $x$ is never disclosed in the process of functionality $F$. For example, an encrypted dataset and a function with secret key will return only the sum of the dataset.

Our goal is to compute the distance between the two encrypted entities. We adopt the inner product encryption (IPE) for this purpose. The IPE is a special case of functional encryption where both secret keys and ciphertexts are associated with vectors. The combination of a secret key $sk_x$ for a vector $x$ and a ciphertexts $ct_y$ for a vector $y$ reveal only $(x \cdot y)$, not anymore about plaintext $x$ or $y$. An inner product encryption scheme is function-hiding if the keys and ciphertexts reveal no additional information about both $x$ and $y$ beyond their inner product. The secret key and ciphertext which are presented as vectors, are modified to compute the distance over the encrypted dataset. The details of encryption and decryption for IPE are described in Appendix.

## Materials and methods

We define the trajectory as a set of pairs of location and time. A location may be either a pair of $(x, y)$ coordinate such as longitude and latitude, or a location ID such as the restaurant name. In this study, we use a location ID rather than a coordinate to check whether persons have visited the same place or not.

**Definition 1** *A trajectory $T^o$ of a moving object o is a set of pairs of locations and time (i.e., $(p_i^o, t_i^o), i \in [1, n]$, with $p_i^o$ is a spatial point, $t_i^o$ is a timestamp, and n is the length of trajectory).*

In an encrypted set of trajectories, all spatial points $p_i$ and timestamps $t_i$ are encrypted. Only the objects' IDs are preserved.

### Dataset

We use the Gowalla Check-in dataset [19] for conducting experiments and demonstrating our system. The detailed description of the dataset is introduced in Table 1. We modified the time

**Table 1. Description of Gowalla dataset.**

| | |
|---|---|
| The number of people | 107,092 |
| The number of locations | 1,280,969 |
| Check-ins | 6,442,892 |
| Time period | 2 year |

period of dataset into one day because the period of two years is not appropriate for contact tracing of COVID-19. The dataset is encrypted with the inner product encryption (IPE) technique [20].

## Numeric decomposition (ND)

Table 2 shows decryption time and encryption time of IPE. Although the number of unique values does not effect on the encryption time, the decryption time gets drastically slower. This is due to the fact that the time complexity of IPE decryption (Appendix) is $O(n^2)$. To improve the performance of decryption, we propose a method of decomposing the location ID and visit time into digits. (e.g. 12345 → 1, 2, 3, 4, 5) As the numeric decomposition (ND) is applied, the decryption time increases linearly as the number of unique values increases (Table 2).

## Privacy-preserving contact tracing workflow

We describe our privacy-preserving contact tracing workflow in Fig 3. First, the trajectories are collected from various sources (e.g. card payment, mobile GPS, and self report). Trajectory data are stored in secured data collectors in an encrypted manner. If a query (i.e., infected person's trajectory data and a time threshold) is given, the query executor retrieves the possible contacts of infected patients and returns the list of contacts with their infection probabilities.

**Table 2. Encryption and decryption time of IPE.**

| Location ID's domain | Encryption time (seconds) | Decryption time (seconds) | ND applied decryption time (seconds) |
|---|---|---|---|
| $1 \sim 10^1$ | 0.1 | 0.01 | 0.01 |
| $1 \sim 10^2$ | 0.1 | 0.1 | 0.02 |
| $1 \sim 10^3$ | 0.1 | 12.6 | 0.03 |
| $1 \sim 10^4$ | 0.1 | 1595.7 | 0.04 |
| $1 \sim 10^5$ | 0.1 | 202957.2 | 0.05 |

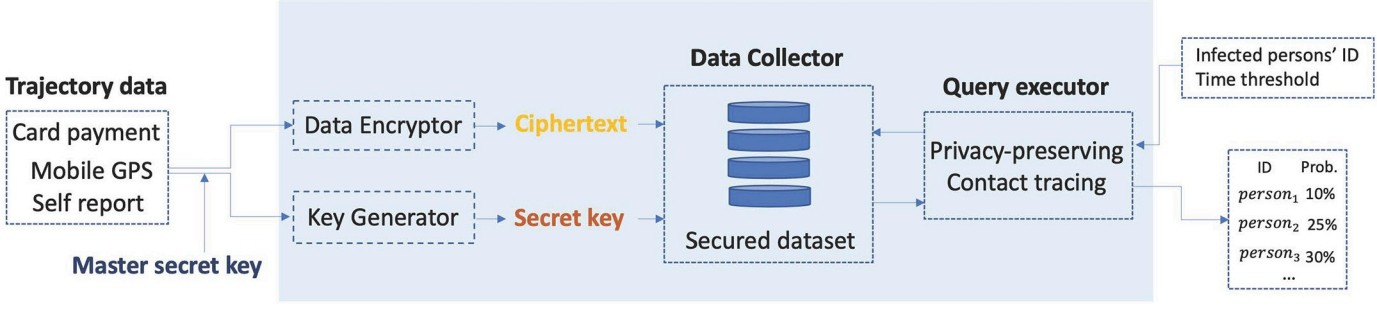

**Fig 3. Privacy-preserving contact tracing workflow.**

## Privacy-preserving contact tracing (PPCT) method

Given a set of encrypted trajectories $T = \{t_1^1, t_2^1, ..., t_n^m\}$, a set of infected persons $I = \{i_1, i_2, ..., i_k\}$, and time threshold $\theta$, a privacy-preserving contact tracing $PPCT$ retrieves a set of persons $U = \{u_1, u_2, ...\}$ whose infection probability $p_{u,i_k}^l$ is greater than zero. The infect probability of a person $u$ from the infected $i_k$ at the location $l_i$ is defined as Eq 1.

$$p_{u,i_k}^l = \begin{cases} min(1, \sum(1 - timegap(u, i_k)/\theta)), & \text{if both } u \text{ and } i_k \text{ visited } l \\ 0, & \text{otherwise} \end{cases} . \quad (1)$$

If the infected person and the suspected person meets many times at the same place, the sum might be larger than one. For example, $\theta$ is given as 30 minutes and the infected person the suspected person meets twice at the same place with ten minutes time gaps. In this case, the infect probability might be computed as four over three. Thus, we add a min function in the equation.

The process of privacy-preserving contact tracing is described in Algorithm 1. As an equality check is required for the spatial dimension, we check location overlap first, and then compute the time gap.

Fig 4 describes an example showing how the PPCT algorithm works. First, card payment data are stored as an encrypted form (**1**). The location is encrypted as a decomposed form. In that, each digit is encrypted as an individual cipher-text. We just notate an alphabet in this example, the real cipher-text is more complicated. Note that the same location 153242 can be encrypted into different cipher-texts because the encryption method is not deterministic. When the PPCT query is given as a set of infected patients' IDs and a temporal threshold, the contact tracing algorithm begins (**2**). Then, the algorithm retrieves the persons who visited the same location with the infected patients (**3**). In this case, persons 2391 and 2013 have been at

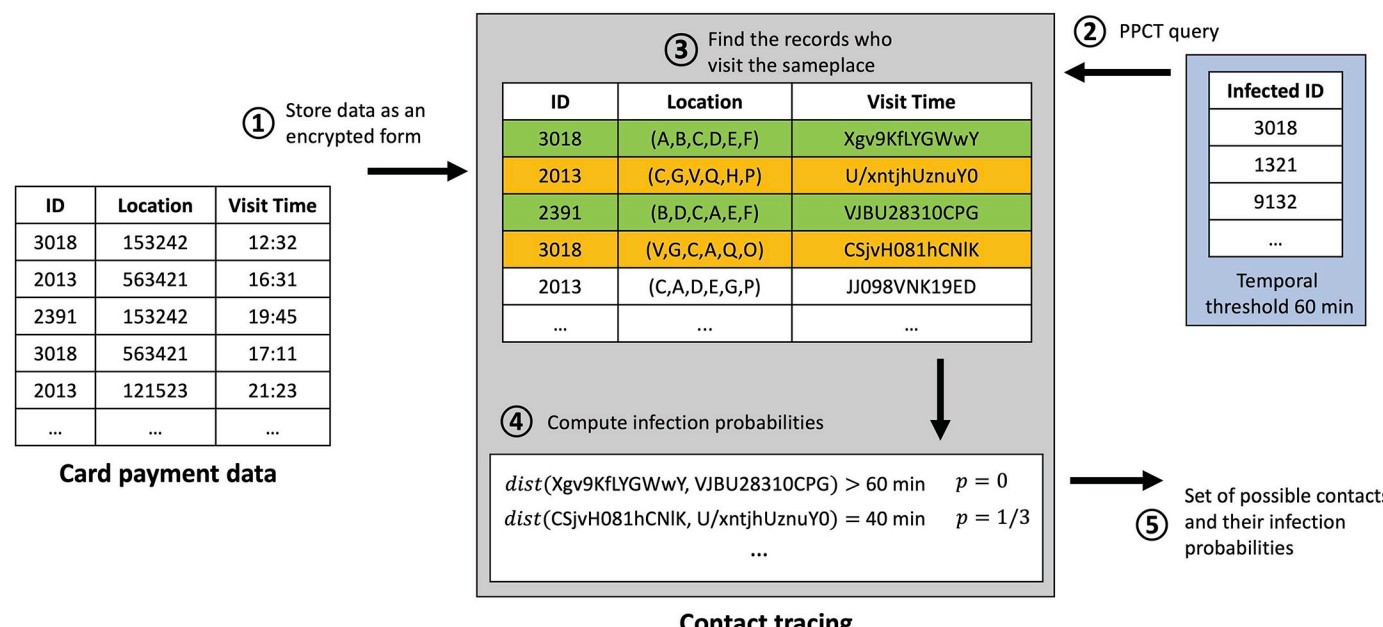

**Fig 4. Example of contact tracing with functional encryption techniques.** The location IDs are decomposed into digits and separately encrypted. The time values are transformed into ordinal numbers and then encrypted. When a set of confirmed IDs is given, the algorithm finds possible contacts by checking the location match and time overlap (within the given threshold 60 min) compared with those of the confirmed cases.

the same place 153242 with the infected patient in 3018. After that, the algorithm computes the infection probabilities for overlapped persons (**4**). The person 2391 was in location 153242 at 19:45 while the infected patient 3018 was in the same location at 12:32. Therefore, their time gap is larger than the temporal threshold (i.e., 60 minutes). The person 2013 was in location 563421 at 16:31 and the infected patient 3018 was in the same location at 17:11. Because their time gap is shorter than the 60 minutes, their infection probability is one third. If the person 2013 and the infected patient 3018 have visited at 563421 several times, the probability will be summed. Surely, the visit time and location are not disclosed in a plain-text form. When the infection probability computation is done for every overlapped visit, the list of possible contacts and their infection probabilities is returned (**5**).

**Algorithm 1**: Privacy-preserving contact tracing PPCT(*ids*, $\theta$)

```
 Input: List of infected persons' ID ids,
      Temporal threshold θ
 Output: Set of suspected persons rs
 Global: Secretkey list skxs, ciphertext list ctys, locations,
visitorsList
        n ← num of records in skxs(or ctys)
        m ← domain size of location
 1 infectedIdx ← []
 2 for i ← 0 to n - 1 do
 3   if skxs[pid][i] in ids then
 4     infectedIdx ← infectedIdx + [i]
 5 newLocations, oldLocations ← findLocations(infectedIdx)
 6 if len(newLocations) ≠ 0 then
 7   for i ← 0 to n - 1 do
 8     for j ← 0 to len(newLocations) - 1 do
 9       index ← newLocations[j]
10       if decrypt(skxs[location][index], ctys[location][i], m²) == 0
then
11          visitorsList[j] ← visitorsList[j] + [i]
12          time_diff ← decrypt(skxs[time][index], ctys[time][i], θ²)
13          if time_diff ≠ -1 and time_diff ≤ θ then
14            rs ← rs + [(locations[j], ctys[pid][i], time_diff)]
15 for i ← 0 to len(oldLocations) - 1 do
16   if len(oldLocations[i]) ≠ 0 then
17     indexes ← oldLocations[i]
18     visitors ← vistorsList[i]
19     for j ← 0 to len(indexes) - 1 do
20       index ← indexes[j]
21       for v ← 0 to len(visitors) - 1 do
22         visitor ← visitors[v]
23         time_diff ← decrypt(skxs[time][index], ctys[time][visitor],
θ²)
24         if time_diff ≠ -1 and time_diff ≤ θ then
25           rs ← rs + [(locations[i], ctys[pid][visitor], time_diff)]
26 return rs
```

First, *PPCT* finds the index of confirmed cases (line 1-4). Then, *PPCT* checks if a person visits the place where infected patients visited. If result of decrypt function is 0, it means that secret key and ciphertext are the same (line 10). We store the visitors who visited the place the infected person has dropped by to skip the decryption next time (line 11). The algorithm finds the person who has contact with the infected person at newLocations (line 6-14). Then, *PPCT* calculates the time gap between the person and the infected (line 12, 23). Lastly, we find the person who has contacted the infected patients at oldLocations (line 15-25).

The *findLocations* function is described in Algorithm 2.

**Algorithm 2**: findLocations(infectedIdx)

```
 Input: infectedIdx
 Output: newLocations, oldLocations
 Global: Secretkey list skxs, ciphertext list ctys, locations,
visitors
        n ← num of records in skxs(or ctys)
        m ← domain size of location
 1 newLocations ← []
 2 oldLocations ← [[], [], ...]
 3 for i ← 0 to len(infectedIdx) − 1 do
 4   index ← infectedIdx[i]
 5   isOldLocation False
 6   for j ← 0 to len(locations) − 1 do
 7     if decrypt(skxs[location][index], locations[j], m²) == 0 then
 8       isOldLocation ← True
 9       oldLocations[j] ← oldLocations[j] + [index]
10       break
11   if isOldLocation == False then
12     locations ← [ctys[location][index]] + locations
13     visitorsList ← [[]] + visitorsList
14     newLocations ← [index] + newLocations
15     oldLocations ← [[]] + oldLocations
16 return newLocations, oldLocations
```

First, we store the index of infected patients who visited place *newLocation* which has never been investigated yet (line 1) and *oldLocations* (line 2). And then, the algorithm checks if the place has been investigated (lines 6-10). If result of decrypt function is 0, it means secret key and ciphertext are the same (line 7). If the infected patients visit investigated place, add index to *oldLocations* (lines 8-10). If the infected patients visit not investigated place, add index to *newLocations* and update *locations*, *visitorList*, and *oldLocations* (lines 11-15).

## Visualization system

We implemented a web-based GUI system to visualize the privacy-preserving contact tracing workflow. The system is implemented with python 3.6.9 version and django 3.0.6 version [21] for the back-end server, and force-directed graph [22] and d3.js [23] for the front-end side. The overview of our visualization system is depicted in Fig 5. The system is composed of three parts (i.e. (A) Infection graph, (B) PPCT Executor, (C) Tracing result table).

In the infection graph (Fig 5(A)), the infected patients and their contacts are described as nodes of graph. Infected patients are colored and the contacts are gray-colored. The edges connecting the nodes signify that they have been at the same location in the time-threshold gap. The moving small dots on the edges represent the infection probability. The higher infection probability is, the faster small dots move along the edges. Users might recognize easily the more suspected contacts by the movement of the small dots. The animation is available in the videos in Appendix.

In PPCT executor (Fig 5(B)), you can choose infected patients and execute the privacy-preserving contact tracing query with a time-threshold. After the query is submitted, the processor redistributes the workload to multi-cores. The processing phase of each core is represented in the progress bars.

In Tracing result table (Fig 5(C)), columns denote the locations and rows denote the persons' IDs. As has been stated above, the locations and the time data are encrypted. An infection probability of a person is depicted as a wave in a circle. For example, in the second row and second column of Fig 5(C), person 90713 was with the gray-colored infected patient (8865). However, the wave shows only 2% of infection probability from the person 90713.

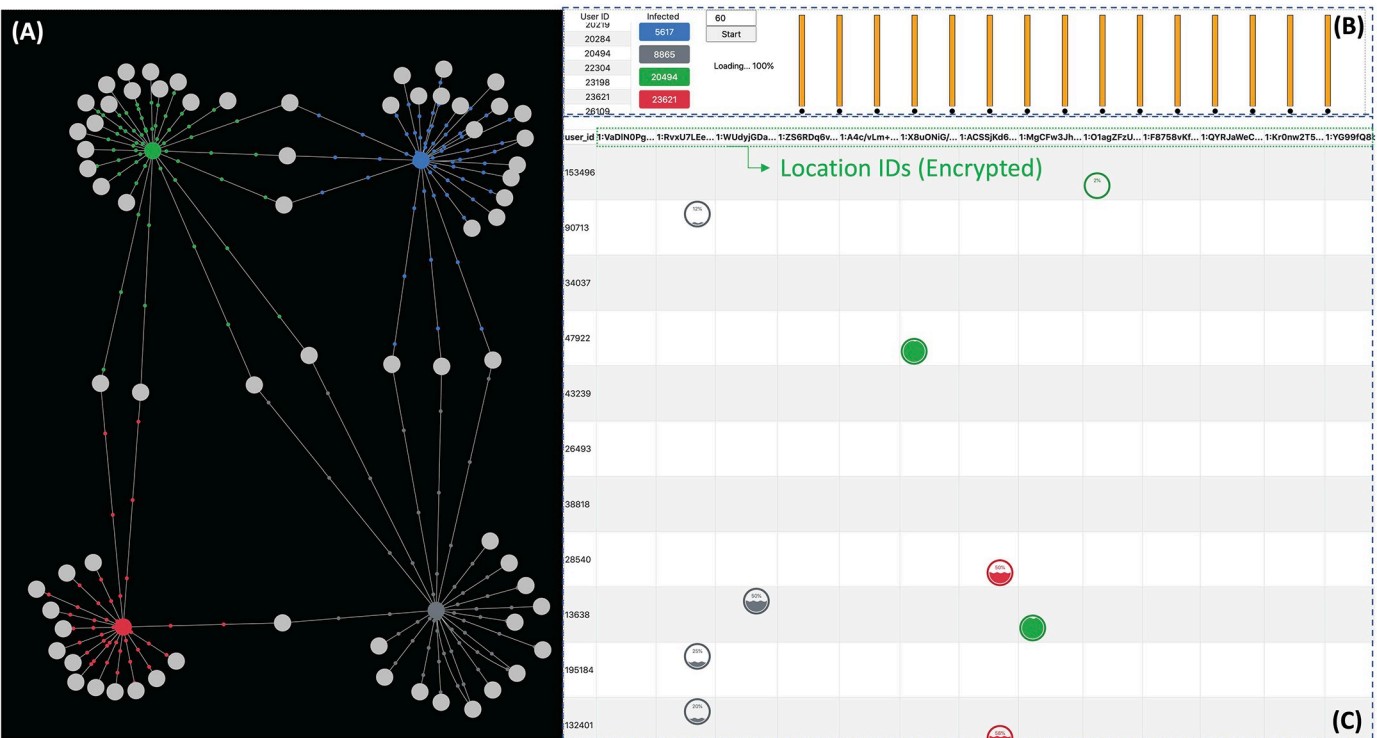

**Fig 5. Visualization system.** (A) Infection graph; Infected patients and their contacts are depicted as a graph. (B) PPCT executor; The progress of privacy-preserving contact tracing is displayed where a bar represents a piece of CPU core. The progress bars of each core are on the right side. (C) Tracing result table; The infection probability of a person is represented as a wave in a circle.

## Results

Fig 6 shows the processing time of PPCT query varying the time threshold and the number of infected people on synthetic (Uniform distribution) and real dataset. The default value $\theta$ is 30 varying the number of infected people, and the default number of infected people is five

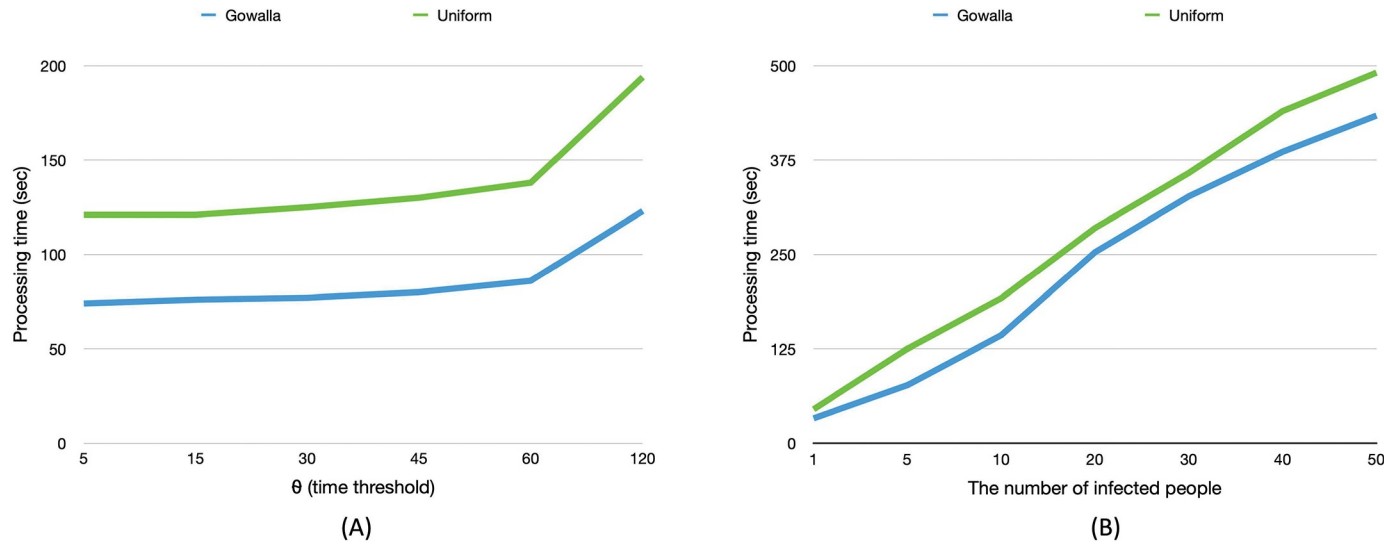

**Fig 6. Query processing time varying the number of infected people and time threshold.**

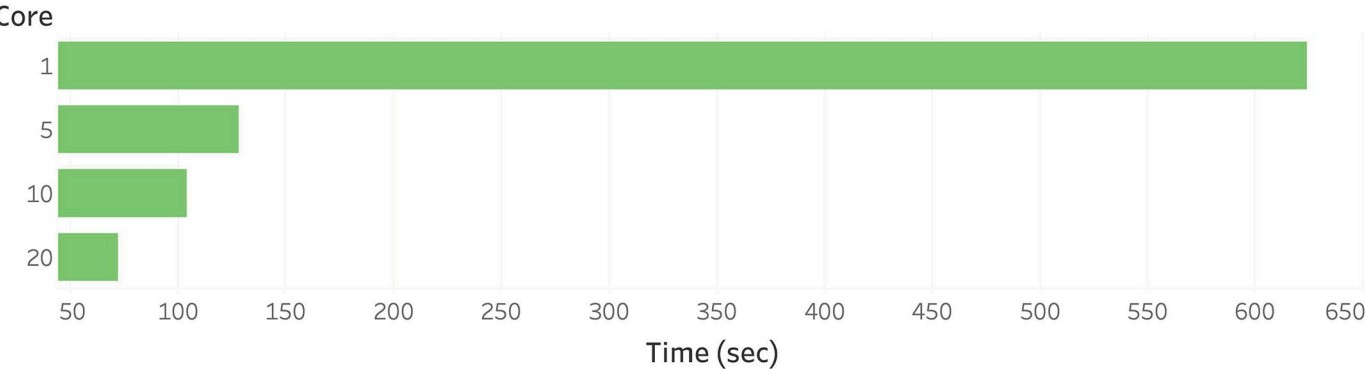

**Fig 7. Query processing time varying the number of multi-cores.**

varying the $\theta$. Fig 6(A) shows that the longer time threshold causes the slower query processing time. This is due to the fact that the longer time threshold causes more suspected people. And the processing time on Uniform distribution tends to longer than the real dataset (Gowalla), it is caused by the fact that people tend to visit the same place. In other words, the number of computations is reduced when a person visits the same place many times. Fig 6(B) denotes the more infected people cause the longer query processing time. The reason for the longer processing time is that the more number of infected people also causes more suspected people to compute the distance. Fig 7 shows the processing time of privacy-preserving contact tracing on 100,000 records varying the number of CPU cores. Twenty-cores improve the performance about ten times faster. As the main workload of the query processing is independently computing the distance between ciphertext and secret key, the encrypted dataset can be simply divided into the number of cores. The CPU-core utilization is monitored to be 100% during the processing regardless of the number of CPU cores. Our system is easily applicable to much bigger dataset if enough CPU cores are provided. As the computation of the decryption is independently executable, not only multi-core processing but also distributed computing can be applied to reduce the query processing time. An interesting future work would be a delegatable version of safe contact tracing query to support multi source environment. To handle multi source datasets safely, each dataset should be encrypted with different keys to prevent attacks from other data sources. We hope a very recent work [24] to be a key reasearch for the delegatable safe contact tracing query.

## Conclusion

In this paper, we proposed a privacy-preserving contact tracing method for COVID-19 based on a functional encryption technique, where the suspected contacts of infected patients can be retrieved without privacy breaches. To efficiently processing, we introduce optimization techniques and parallel processing. A visualization system is also designed to visualize the workflow of contact tracing.

## Supporting information

**S1 Video. Video for introducing our visualization.** Available at https://youtu.be/5_2TR3l-Fhw.
(TXT)

**S1 Appendix. IPE encryption / decryption algorithm.**
(PDF)

## Author Contributions

**Conceptualization:** Hyubjin Lee.

**Methodology:** Wooil Kim, Hyubjin Lee.

**Project administration:** Wooil Kim, Yon Dohn Chung.

**Software:** Hyubjin Lee.

**Visualization:** Wooil Kim.

**Writing – original draft:** Wooil Kim.

**Writing – review & editing:** Yon Dohn Chung.

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
