## [Decision Letter · Decision Letter 0]

29 Jul 2020

PONE-D-20-19705

Safe contact tracing for COVID-19: A method without privacy breach using functional encryption techniques based-on spatio-temporal trajectory data

PLOS ONE

Dear Dr. Chung,

Thank you for submitting your manuscript to PLOS ONE. After careful consideration, we feel that it has merit but does not fully meet PLOS ONE’s publication criteria as it currently stands. Therefore, we invite you to submit a revised version of the manuscript that addresses the points raised during the review process.

We look forward to receiving your revised manuscript.

Kind regards,

He Debiao

Academic Editor

PLOS ONE

Journal Requirements:

Reviewers' comments:

Reviewer's Responses to Questions

**Comments to the Author**

1. Is the manuscript technically sound, and do the data support the conclusions?

Reviewer #1: Yes

Reviewer #2: Yes

2. Has the statistical analysis been performed appropriately and rigorously? 

Reviewer #1: No

Reviewer #2: Yes

3. Have the authors made all data underlying the findings in their manuscript fully available?

Reviewer #1: Yes

Reviewer #2: Yes

4. Is the manuscript presented in an intelligible fashion and written in standard English?

Reviewer #1: Yes

Reviewer #2: Yes

5. Review Comments to the Author

Reviewer #1: This work focuses on a privacy-preserving contact tracing application based on trajectory data of potentially infected individuals. The motivation of the paper is laid out well in that without preserving the privacy of individuals, one may be hindering the containment efforts for the global pandemic. Privacy-preserving contact tracing (PPCT) method uses functional encryption which allows for a so-called spatio-temporal similarity search (at the same place in similar times) without exposing the information of its users and potentially infected individuals. There are minor grammatical errors which can be revisited later. Some of the major and minor issues can be seen below.

Major:

- Alg 1 is the core contribution and its psuedocode and explanation can be substantially improved. (e.g., what exactly is meetFlag? or controls with breaks and continue's can be avoided, or differences between two append/add operations [step 15&31] can be shown)

- A better explanation for infection probability (Eq 1) can be provided

- Provided that temporal domain (and the associated search) is 1-dimensional while spatial is 2, it is often more efficient to filter using time and refine with a spatial search. A clarification on why spatial search (location overlap) was performed first can be helpful.

- The time threshold (theta), can be instrumental in eliminating the resulting candidates. An explanation on how it can be used to speed up the search or further refine the test set can also be helpful. Alternatively, it can be shown via experiments that such parameter does or does not impact the results.

- More experimentation with either real-life or synthetic datasets can be helpful as well in that what would be the impact of higher number of infections, how does the infection probability (or its parameters) impact the performance of the search?

Minor:

- In Fig 1, line charts instead of area charts can be used, as it is not clear whether the charts are stacked or not. One could preferably use the log scale including the more current data.

- Pg. 2-Ln. 22-24. Steps can be rearranged to follow the progression (as in 3->4a->4b) or Fig 2 can be rearranged, as in Step 4a->3, Step 3->4.a.

- It is a bit confusing to use location as a pair of spatial point and time, since location is often used in purely spatial context.

Reviewer #2: In this paper, the authors proposed a privacy-preserving contact tracing method based on spatio-temporal trajectory which can be practically used in many quarantine systems. In addition, a system is designed to visualize the contact tracing workflow. The paper has significant contributions amid the Covid-19 pandemic. I have a few Minor concerns given below:

1. On page 8, line#32 please explain how the travel route data expose the privacy or what information the travel route data contains.

2. Please revise this sentence "No consideration of privacy even decreases the efficiency of epidemic controls because the suspected contacts might refuse to take infection tests because they are afraid of privacy breach". because is used frequently

3. Please state what is a domain ?

4. Line# 181 , Line#182 please provide specifications of the back-end server and version of python.

5. Line#191 The animation is available on the videos in Appendix. Please revise " The animation is available (in) the videos.

6. Add another graph showing the cpu utilization comparison with varying time for the cores.

7. In references, Please write the accessed date i.e. when you are refereing to a website. e.g. Reference 20, 21 and 22 are without an accessed date.

6. PLOS authors have the option to publish the peer review history of their article (what does this mean?). If published, this will include your full peer review and any attached files.

Reviewer #1: No

Reviewer #2: **Yes: **JEHAD ALI

---

## [Author Response · Author response to Decision Letter 0]

13 Aug 2020

The authors would like to thank to the editor He Debiao, and reviewer JEHAD ALI, and the anonymous reviewer.

---

## [Decision Letter · Decision Letter 1]

10 Nov 2020

Safe contact tracing for COVID-19: A method without privacy breach using functional encryption techniques based-on spatio-temporal trajectory data

PONE-D-20-19705R1

Dear Dr. Chung,

We’re pleased to inform you that your manuscript has been judged scientifically suitable for publication and will be formally accepted for publication once it meets all outstanding technical requirements.

Kind regards,

He Debiao

Academic Editor

PLOS ONE

Additional Editor Comments (optional):

Reviewers' comments:

Reviewer's Responses to Questions

**Comments to the Author**

1. If the authors have adequately addressed your comments raised in a previous round of review and you feel that this manuscript is now acceptable for publication, you may indicate that here to bypass the “Comments to the Author” section, enter your conflict of interest statement in the “Confidential to Editor” section, and submit your "Accept" recommendation.

Reviewer #2: All comments have been addressed

2. Is the manuscript technically sound, and do the data support the conclusions?

Reviewer #2: Yes

3. Has the statistical analysis been performed appropriately and rigorously? 

Reviewer #2: Yes

4. Have the authors made all data underlying the findings in their manuscript fully available?

Reviewer #2: Yes

5. Is the manuscript presented in an intelligible fashion and written in standard English?

Reviewer #2: Yes

6. Review Comments to the Author

Reviewer #2: The authors have addressed all my concerns and questions in the revised version of the manuscript. In my opinion, the manuscript is suitable for publication.

7. PLOS authors have the option to publish the peer review history of their article (what does this mean?). If published, this will include your full peer review and any attached files.

Reviewer #2: **Yes: **DR JEHAD ALI

---

## [Editor Report · Acceptance letter]

3 Dec 2020

PONE-D-20-19705R1 

Safe contact tracing for COVID-19: A method without privacy breach using functional encryption techniques based-on spatio-temporal trajectory data 

Dear Dr. Chung:

I'm pleased to inform you that your manuscript has been deemed suitable for publication in PLOS ONE. Congratulations! Your manuscript is now with our production department. 

Kind regards, 

on behalf of

Dr. He Debiao 

Academic Editor

PLOS ONE